# Discovery and characterization of a novel family of prokaryotic nanocompartments involved in sulfur metabolism

Robert J Nichols[1†], Benjamin LaFrance[1†], Naiya R Phillips[1], Devon R Radford[2], Luke M Oltrogge[1], Luis E Valentin-Alvarado[3], Amanda J Bischoff[4], Eva Nogales[1,5,6,7,8]*, David F Savage[1]*

[1]Department of Molecular and Cell Biology, University of California, Berkeley, Berkeley, United States; [2]Department of Molecular Genetics, University of Toronto, Toronto, Canada; [3]Department of Plant and Microbial Biology, University of California, Berkeley, Berkeley, United States; [4]Department of Chemistry, University of California Berkeley, Berkeley, United States; [5]Howard Hughes Medical Institute, University of California, Berkeley, Berkeley, United States; [6]Molecular Biophysics and Integrated Bioimaging Division, Lawrence Berkeley National Laboratory, Berkeley, United States; [7]California Institute for Quantitative Biosciences (QB3), University of California, Berkeley, Berkeley, United States; [8]Molecular Biophysics and Integrated Bio-Imaging Division, Lawrence Berkeley National Laboratory, Berkeley, United States

**Abstract** Prokaryotic nanocompartments, also known as encapsulins, are a recently discovered proteinaceous organelle-like compartment in prokaryotes that compartmentalize cargo enzymes. While initial studies have begun to elucidate the structure and physiological roles of encapsulins, bioinformatic evidence suggests that a great diversity of encapsulin nanocompartments remains unexplored. Here, we describe a novel encapsulin in the freshwater cyanobacterium *Synechococcus elongatus* PCC 7942. This nanocompartment is upregulated upon sulfate starvation and encapsulates a cysteine desulfurase enzyme via an N-terminal targeting sequence. Using cryo-electron microscopy, we have determined the structure of the nanocompartment complex to 2.2 Å resolution. Lastly, biochemical characterization of the complex demonstrated that the activity of the cysteine desulfurase is enhanced upon encapsulation. Taken together, our discovery, structural analysis, and enzymatic characterization of this prokaryotic nanocompartment provide a foundation for future studies seeking to understand the physiological role of this encapsulin in various bacteria.

**\*For correspondence:**
enogales@lbl.gov (EN);
savage@berkeley.edu (DFS)

†These authors contributed equally to this work

**Competing interests:** The authors declare that no competing interests exist.

## Introduction

Subcellular compartmentalization is an essential strategy used by cells to facilitate metabolic pathways that are incompatible with the rest of the cytosol. Contrary to common misconceptions that organelles are exclusive to eukaryotes, even prokaryotes partition metabolic pathways into unique chemical environments using subcellular compartments (*Grant et al., 2018*). For example, studies of the bacterial microcompartments called carboxysomes have shown how the complex sequesters the enzyme rubisco and facilitates substrate channeling by increasing the local concentration of $CO_2$ (*Oltrogge et al., 2020*; *Mangan et al., 2016*; *Kerfeld et al., 2018*). In addition to modulating cargo activity, compartmentalization can also provide a means of sequestering toxic intermediates of metabolic pathways from the rest of the cell. For example, the propane-diol utilization (PDU) microcompartment sequesters a cytotoxic aldehyde intermediate from the cytoplasm and allows it to be

subsequently converted by downstream, compartmentalized enzymes to efficiently generate the end products of the pathway (*Sampson and Bobik, 2008*; *Crowley et al., 2010*; *Chowdhury et al., 2015*; *Kerfeld et al., 2018*).

Recently, another class of protein-bounded compartments, known as prokaryotic nanocompartments, has been discovered (*Sutter et al., 2008*). These nanocompartments, also called encapsulins, are smaller and less complex than microcompartments. They are typically found as a two-gene system which encodes a shell protein that self-assembles into an icosahedral capsid-like compartment, and a cargo protein that becomes encapsulated by the shell through a targeting peptide sequence (*Giessen, 2016*; *Nichols et al., 2017*). Many functionally diverse cargo proteins have been found to be associated with encapsulins, including ferritin-like proteins (FLP), iron mineralizing encapsulin cargo from firmicutes (IMEF), DyP-type peroxidases, and hydroxylamine oxidoreductase (HAO) (*Sutter et al., 2008*; *Giessen et al., 2019*; *Giessen and Silver, 2017*; *Xing et al., 2020*). The precise physiological role of these compartments remains elusive except for a few instances. Notably, the DyP-containing encapsulins from *Myxococcus xanthus* have been implicated in nutrient starvation and oxidative stress responses (*Kim et al., 2009*; *Kim et al., 2019*; *McHugh et al., 2014*). The FLP and IMEF containing encapsulins appear to be involved in iron storage and mitigation of toxic reactive oxygen species products of the Fenton reaction due to free iron during oxidative stress (*Giessen et al., 2019*; *Giessen and Silver, 2017*; *He et al., 2016*). Encapsulins are also thought to be integral to highly specialized metabolism, such as that found in anammox bacteria, in which the HAO cargo has been hypothesized to reduce a cytotoxic hydroxylamine metabolic intermediate (*Giessen and Silver, 2017*; *Kartal et al., 2013*; *Xing et al., 2020*).

Based on the evidence accumulated thus far, it appears that encapsulins play diverse physiological roles. Despite this diversity, encapsulation of redox reactions is a recurring theme (*Nichols et al., 2017*). Thus far, study of this expansive repertoire of encapsulins has been limited to the homologs of closely related compartment shell proteins. Here, we describe a new family of nanocompartment systems that are evolutionarily distinct from those previously reported. Specifically, we implicated a role for this encapsulin family in the sulfur starvation response. Further, we have identified a unique cysteine desulfurase cargo enzyme and elucidated an N-terminal encapsulation targeting sequence that is necessary and sufficient for compartmentalization. Finally, we report a high-resolution structure (2.2 Å) of the complex and identify the cargo binding site within the compartment. This structure greatly informs our model for the biochemical function of this novel organelle-like compartment.

## Results

### A novel family of predicted prokaryotic nanocompartments is widespread throughout bacterial phyla

A unifying feature of the encapsulin nanocompartments is the shared HK97 phage-like fold of the shell protein (*Giessen and Silver, 2017*; *Nichols et al., 2017*). Bacteriophages belonging to the order Caudovirales also possess a major capsid protein that is structurally homologous to the HK97 fold. Given the shared homology of encapsulins and Caudovirales capsid proteins, an evolutionary relationship between the two has been proposed (*Koonin and Krupovic, 2018*; *Krupovic et al., 2019*; *Krupovic and Koonin, 2017*), and the existence of other bacteriophage-related nanocompartments beyond the close relatives of known encapsulins has been postulated (*Radford, 2015*; *Giessen, 2016*; *Nichols et al., 2017*). Recently, a bioinformatic study explored this possibility by searching prokaryotic genomes for phage capsid genes that are unlikely to be functional phages, but may actually be putative encapsulins (*Radford, 2015*). This search suggested that previously published encapsulins, hereafter referred to as Family 1 encapsulins, comprise a minor fraction of all encapsulin systems. Here, we report the first characterization of a novel encapsulin family, which we term Family 2. This novel family is even more prevalent than Family 1 encapsulins (*Supplementary file 1*) and is present in many model organisms. Despite the prevalence of Family 2 encapsulins, the experimental characterization of this family as a prokaryotic nanocompartment has never been explored.

Phylogenetic analysis of the encapsulin shell proteins revealed that Family 2 encapsulins are distinct from the previously published Family 1 systems (*Figure 1A*). Family 2 further divides into what

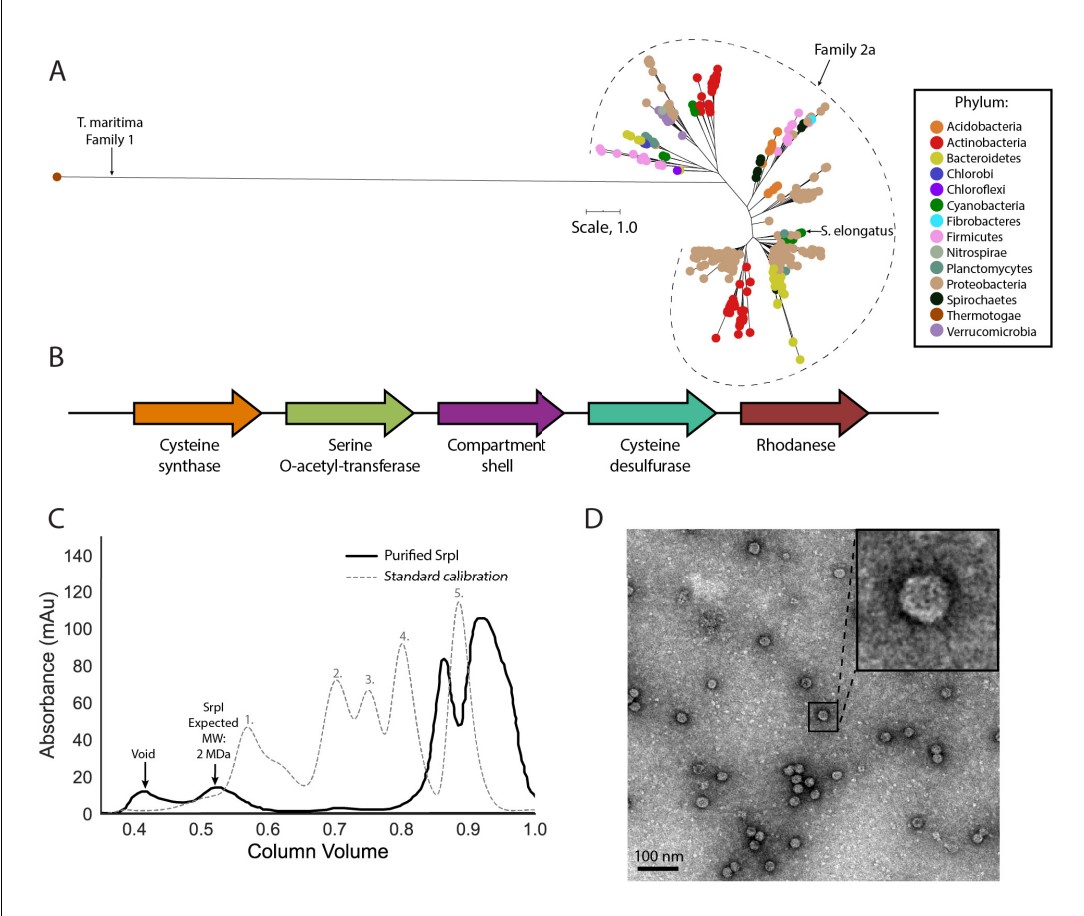

**Figure 1.** SrpI is a bacterial nanocompartment that is widespread throughout bacterial phyla and found neighboring sulfur metabolism genes. (**A**) Maximum-likelihood phylogenetic tree of Family 2a encapsulin shell proteins using the *T. maritima* Family 1 encapsulin shell protein (WP_004080898.1) as a Family 1 representative. Scale bar, one substitution per site. (**B**) Genomic neighborhood of the Family 2a encapsulin shell gene from *S. elongatus* PCC 7942. (**C**) Size exclusion-chromatogram of purified SrpI shell protein (black solid line) and Bio-Rad gel filtration calibration standard (gray dashed line) using a Superose 6 Increase column (GE Life Sciences). Expected molecular weight was determined using Bio-Rad gel filtration calibration standard: 1. Thyroglobin (670 kDa) 2. γ-globulin (158 kDa) 3. Ovalbumin (44 kDa) 3. Myoglobin (17 kDa) Vitamin B12 (1.4 kDa). (**D**) Negative stain TEM micrograph of resulting SrpI encapsulin-containing fraction post size-exclusion chromatography. Scale bar, 100 nm.

The online version of this article includes the following figure supplement(s) for figure 1:

**Figure supplement 1.** Family 2 encapsulins can be divided into two phylogenetically distinct subfamilies.

**Figure supplement 2.** Family 2b shell genes neighbor 2-methylisoborneol synthase or polyprenyl diphosphate synthase.

**Figure supplement 3.** The Family 2 encapsulin, SrpI, forms a high-molecular-weight complex similar to the Family 1 encapsulin from *T. maritima*.

**Figure supplement 4.** Size distribution of SrpI encapsulin.

we propose as two distinct subfamilies, Family 2a and Family 2b. The subfamilies can be distinguished from each other by their phylogenetic clustering (*Figure 1—figure supplement 1*). Notably, the two subfamilies are found in distinct genomic contexts. Family 2a is found adjacent to sulfur metabolism genes whereas Family 2b neighbors genes involved in terpenoid synthesis (*Figure 1B*; *Figure 1—figure supplement 2*). Most prevalent among the Family 2a subfamily was the co-occurrence of a neighboring cysteine desulfurase gene, while the Family 2b shell genes were found to most often neighbor a polyprenyl diphosphate synthase gene (*Supplementary file 2* and *3*). The individual subfamilies can also be defined by the CRP/FNR cyclic nucleotide-binding domain that is predominantly found in the Family 2b shell sequences but not Family 2a. Our focus for the remainder of the paper will be on the Family 2a subfamily, which is widespread and found distributed in a polyphyletic fashion throughout 13 bacterial phyla (*Figure 1A*).

One such occurrence of Family 2a is in the model cyanobacterium *Synechococcus elongatus* PCC 7942 (henceforth *S. elongatus*) and we sought to validate whether the predicted encapsulin shell

gene (Synpcc7942_B2662, SrpI) was indeed part of a nanocompartment complex. Expression of the shell gene in *Escherichia coli* BL21 (DE3) cells, followed by purification and size-exclusion chromatography, revealed that the protein eluted with an estimated molecular weight of ≈2 MDa (*Figure 1C*), the typical size for many previously characterized encapsulins (*Cassidy-Amstutz et al., 2016*; *Snijder et al., 2016*). Consistent with the previously-characterized Family 1 encapsulin from *Thermotoga maritima*, a high-molecular-weight band was detected with SDS-PAGE analysis for non-heat denatured samples. Boiling the sample yielded a band at 35 kDa, the expected weight of the monomeric shell protein (*Figure 1—figure supplement 3*; *Cassidy-Amstutz et al., 2016*). Negative stain transmission electron microscopy (TEM) of the purified sample indicated the complex forms a nanocompartment with an average diameter of 25 ± 1 nm (n = 180) (*Figure 1D*, *Figure 1—figure supplement 4*).

## SrpI encapsulin is upregulated under sulfur starvation and hosts a cysteine desulfurase cargo protein

Previous work by Nicholson and colleagues in *S. elongatus* demonstrated that the encapsulin shell gene (Synpcc7942_B2662) is one of many whose mRNA expression level is upregulated upon sulfur starvation (*Nicholson et al., 1995*; *Nicholson and Laudenbach, 1995*). Thus, this gene, which is found on a plasmid encoding many sulfur-related genes, was termed SrpI for *S*ulfur *r*egulated *p*lasmid-encoded gene-I (*Chen et al., 2008*; *Nicholson and Laudenbach, 1995*). In order to validate this result at the protein level, we sulfur starved wild-type *S. elongatus* cells for the duration of a 48 hr time-course to detect the upregulation of the nanocompartment and, potentially, identify additional cargo via mass spectrometry.

Consistent with previous studies of sulfur starvation in cyanobacteria, we observed the expected chlorosis phenotype due to the degradation of phycobilisomes (*Collier and Grossman, 1992*; *Cameron and Pakrasi, 2010*). Chlorosis was confirmed by loss of phycocyanin absorbance at 620 nm (*Figure 2A*, *Figure 2—figure supplement 1*). During this time-course, SDS-PAGE analysis of cell lysates also indicated upregulation of a high-molecular-weight complex (*Figure 2B*). Bands were excised, proteolytically digested, and analyzed via liquid chromatography–mass spectrometry. After 48 hr of sulfur starvation, the top hits, as determined by total spectral counts, were the putative encapsulin shell protein, SrpI, and the product of the neighboring gene (Synpcc7942_B2661), a cysteine desulfurase (*Figure 2C*). While we were able to detect upregulation of SrpI and cysteine desulfurase during sulfur starvation, knockout mutants for these genes in *S. elongatus* PCC 7942 did not yield a growth defect phenotype under nutrient replete nor sulfur starvation conditions (*Figure 2—figure supplement 2*). Taken together, these results suggest that the cysteine desulfurase, which neighbors the SrpI shell gene, is the encapsulated cargo protein (*Supplementary file 2*, *Nichols et al., 2017*).

## A disordered N-terminal domain targets cargo for SrpI encapsulation *in vivo*

Sequence alignment of the five cysteine desulfurases found in the *S. elongatus* genome revealed that the SrpI-associated cysteine desulfurase (Synpcc7942_B2661), hereafter named CyD, possesses a unique N-terminal domain in addition to the canonical cysteine desulfurase domain (*Figure 3A*, *Figure 3—figure supplement 1*). This N-terminal domain is shared by cysteine desulfurases found adjacent to SrpI homologs in species possessing this encapsulin system. Structural prediction using the primary sequence of the SrpI-associated CyD revealed that the N-terminal domain (NTD) is highly disordered (*Figure 3A*). Intrinsically disordered domains are known to evolve rapidly, preserving bulk chemical characteristics even as the sequence diverges greatly (*Moesa et al., 2012*; *Varadi et al., 2015*). Accordingly, sequence conservation throughout the N-terminal domain is sparse. However, two motifs, 'LARLANEFFS' and 'AASPYYFLDG', can be found in most SrpI-associated CyD sequences (*Figure 3—figure supplement 2*; *Figure 3B*).

We next sought to confirm that CyD is the cargo protein by using the N-terminal domain to target heterologous cargo to the compartment. To identify the minimal sequence necessary for encapsulation, truncated sequences of the CyD cargo gene were fused to the superfolder green fluorescent protein variant (sfGFP) and co-expressed with the shell protein in *E. coli*. This same approach has been applied to identify targeting sequences for the Family 1 encapsulins (*Cassidy-*

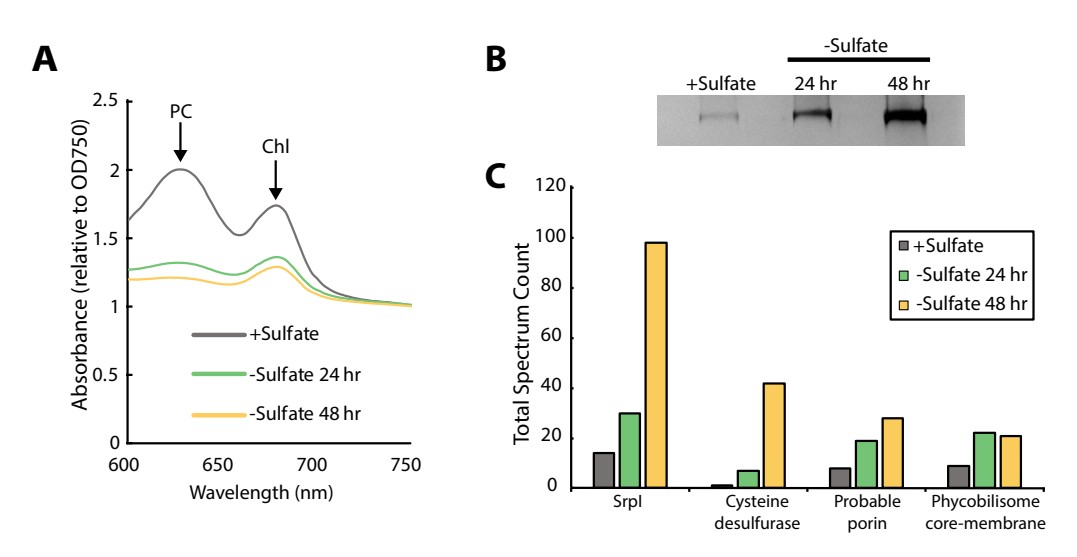

**Figure 2.** SrpI encapsulin is upregulated in *S. elongatus* upon sulfate starvation. (**A**) Absorbance spectra of *S. elongatus* liquid cultures under nutrient-replete conditions (+Sulfate) and sulfur starvation (-Sulfate) for 24 and 48 hr. Absorbance maxima of phycocyanin (PC) at 620 nm and chlorophyll (Chl) at 680 nm are indicated. Absorbance spectra are normalized to the same optical density at 750 nm. (**B**) Non-denaturing SDS-PAGE analysis of lysates from nutrient-replete and sulfur starved *S. elongatus* cultures visualized by silver stain. Inputs were normalized using absorbance at 280 nm. (**C**) Liquid chromatography-mass spectrometry of excised high-molecular-weight bands from SDS-PAGE analysis. Top protein hits from each condition are represented by total spectrum counts. Total spectrum count of all identified proteins can be found in *Supplementary file 5*.

The online version of this article includes the following figure supplement(s) for figure 2:

**Figure supplement 1.** Chlorosis phenotype of sulfur-starved *S. elongatus* PCC 7942.

**Figure supplement 2.** Growth curves of *S. elongatus* PCC 7942 mutants under sulfur replete and sulfur starvation conditions.

**Figure supplement 3.** Non-denaturing SDS-PAGE analysis of lysates from nutrient-replete and sulfur starved *S. elongatus* cultures visualized by silver stain.

*Amstutz et al., 2016*). Examination of these expressed constructs via SDS-PAGE and Coomassie stain showed that all constructs formed the nanocompartment complex, as indicated by the presence of the signature high-molecular-weight band that also served as a loading control. Encapsulation of the heterologous sfGFP-fusion cargo was assayed by measuring GFP fluorescence of the high-molecular-weight band (*Figure 3C*). The entire 225 amino acid N-terminal domain from CyD fused to sfGFP (CyD 1–225-sfGFP) yielded the highest loading signal. Targeting with the first 100 amino acids of the NTD also functioned, albeit not as efficiently as the full N-terminus. The entire cysteine desulfurase fused to sfGFP was also encapsulated, yet again not as well as the 225-NTD. This discrepancy may be due to steric hindrance resulting in fewer copies of the larger full-length CyD-sfGFP construct physically packed inside the compartment. Lastly, CyD 155–183-sfGFP, containing the conserved motif of 'AASPYYFLDG', was not sufficient to sequester cargo within the compartment, nor was the non-tagged sfGFP construct.

## The CyD NTD is necessary and sufficient for loading heterologous cargo *in vitro*

Prior work demonstrated that it is possible to assay cargo loading by disassembling the shell protein with a chaotrope, such as guanidine hydrochloride (GuHCl), and re-folding the shell protein in the presence of cargo protein (*Cassidy-Amstutz et al., 2016*). In this manner, we can control the amount of cargo protein and ensure that loading is due to the targeting sequence rather than mass-action. The shell protein was purified, unfolded in GuHCl, and then refolded by dilution in the presence of purified 225NTD-sfGFP or untagged sfGFP. After refolding and concentration of the sample, the loaded compartment fraction was separated from un-loaded cargo via size exclusion chromatography (*Figure 4—figure supplement 1*). Again, encapsulation was assayed via SDS-PAGE analysis. Only the 225NTD-sfGFP construct displayed GFP fluorescence in the high-molecular-weight band, indicative of sfGFP loading (*Figure 4A*). Furthermore, analysis of the compartment and cargo

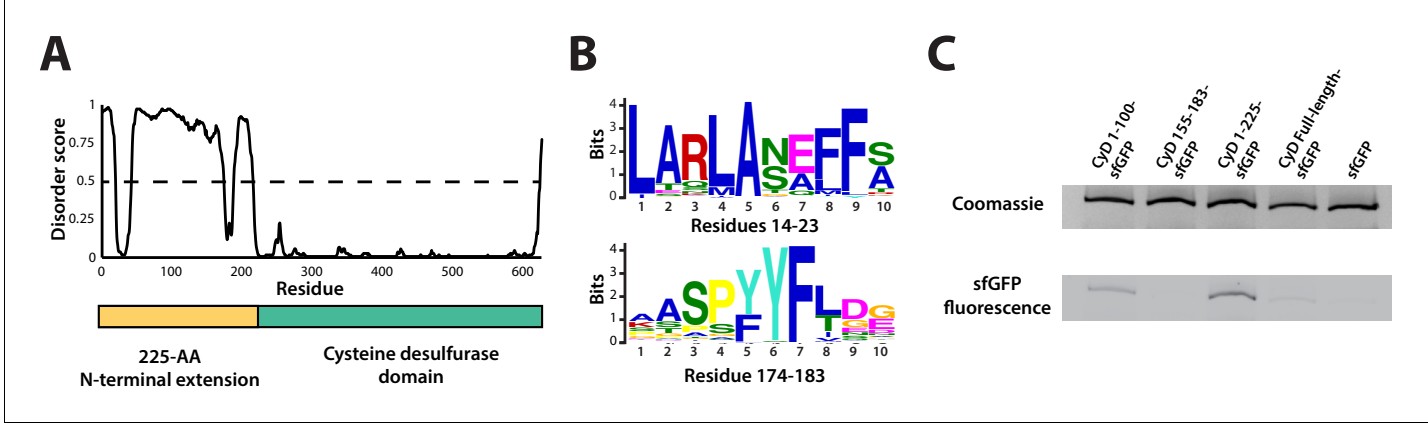

**Figure 3.** An N-terminal signal sequence directs cargo loading *in vivo*. (**A**) Domain organization of cysteine desulfurase (CyD; Synpcc7942_B2661) and the predicted disorder scores calculated using DISOPRED3. CyD can be split into two domains – a highly disordered N-terminal domain and an ordered cysteine desulfurase domain. (**B**) Sequence WebLogos of conserved motifs found within the N-terminal domain of CyD calculated using the MEME suite motif discovery server. (**C**) SDS-PAGE analysis of CyD constructs fused to sfGFP. Loading of fusion cargo and untagged-sfGFP control was determined by fluorescence of the nanocompartment band prior to Coomassie staining.

The online version of this article includes the following figure supplement(s) for figure 3:

**Figure supplement 1.** SrpI-associated cysteine desulfurase, CyD, possesses a unique N-terminal domain that is absent in the four other cysteine desulfurase genes in the *S. elongatus* PCC 7942 genome.

**Figure supplement 2.** Sequence conservation of the F2A encapsulin-associated cysteine desulfurase is sparse throughout the disordered N-terminal domain.

**Figure supplement 3.** Full gel from Coomassie stain panel of *Figure 3C*.

**Figure supplement 4.** Full gel from sfGFP fluorescence panel of *Figure 3C*.

fractions using denaturing SDS-PAGE showed the presence of cargo protein in the compartment fraction only for the 225NTD-sfGFP construct, whereas the presence of sfGFP lacking the NTD was only found in the cargo fraction (*Figure 4B*).

We were also able to load the full-length cysteine desulfurase *in vitro* using the same procedure. We observed encapsulation of the native cargo as indicated by the co-elution of cargo in the compartment fraction (*Figure 4C*). Lastly, we found that the disordered NTD is essential for cargo loading. A mutant CyD lacking the entire N-terminal domain (ΔNTD-CyD) was not measurably encapsulated, as evidenced by separate elution of compartment and truncated cargo (*Figure 4C*).

## Visualization of cargo density by transmission electron microscopy

To further characterize the interaction between SrpI encapsulin and the CyD cargo, we first used negative stain TEM to understand the spatial organization of the CyD cargo within the nanocompartment. In order to obtain sufficient yield for analysis, SrpI and CyD were expressed and purified separately from *E. coli*. Holo-SrpI (CyD loaded) was generated by denaturation of the purified SrpI shell protein and re-folding the shell protein in the presence of CyD as described in the previous section. Apo-SrpI encapsulin (no cargo) was obtained by refolding the SrpI shell in absence of CyD cargo protein. Even in raw micrographs, a clear difference in density on the shell interior was observed when comparing the apo-SrpI encapsulin (no cargo) to the holo-SrpI encapsulin (CyD loaded) (*Figure 5A,C*). The apo-SrpI encapsulin structure revealed the shell without any internal cargo density, with modest features clearly visible as expected from a ≈ 20 Å negative stain reconstruction (*Figure 5B*, Materials and methods). The lack of density within the interior of the apo-SrpI capsid also demonstrates that the shell is permissive to uranyl formate stain, which would also allow definition of internal features for the cargo-loaded sample. For the cargo-loaded structure, clear density for the CyD exists at one of the sites of threefold symmetry within the complex (*Figure 5D*). A difference map, created by subtracting apo-SrpI from the holo-SrpI structure, revealed density corresponding to 1–2 copies of the cysteine desulfurase (*Figure 5E*).

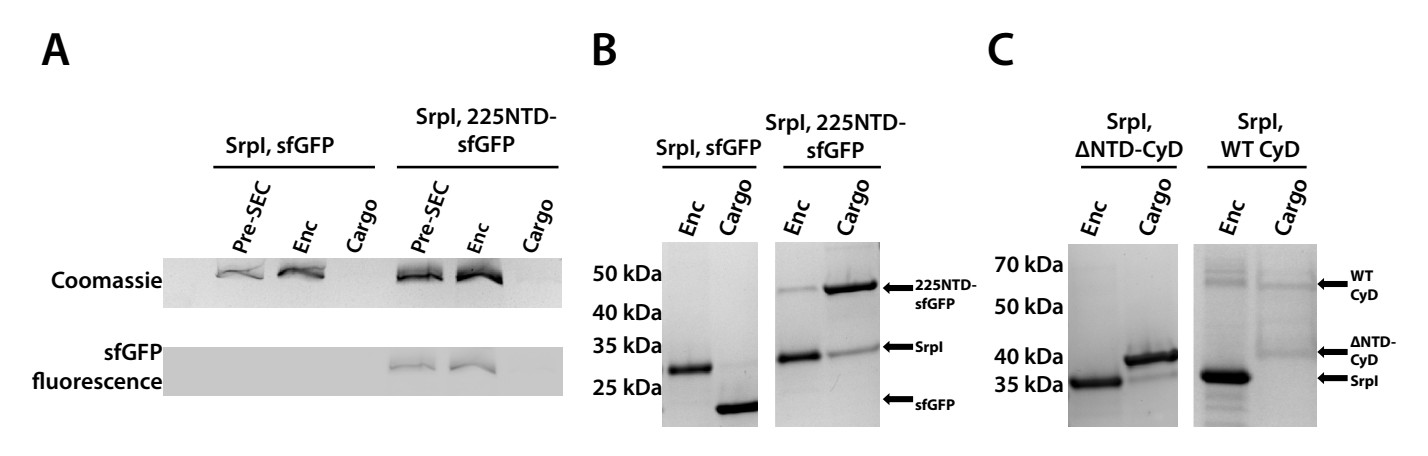

**Figure 4.** The 225-NTD of CyD is necessary and sufficient for cargo loading *in vitro*. (**A**) Non-denaturing SDS-PAGE of sfGFP or the CyD N-terminal domain-sfGFP fusion (225NTD-sfGFP) loaded *in vitro* into SrpI encapsulin. sfGFP fluorescence followed by Coomassie staining of the encapsulin (Enc), cargo, and pre size-exclusion chromatography (Pre-SEC) fractions was performed to determine cargo loading and presence of nanocompartment, respectively. (**B**) Denaturing SDS-PAGE of *in vitro* loaded sfGFP and 225NTD-sfGFP samples to determine presence of SrpI shell monomer (35 kDa), sfGFP (27 kDa), and 225NTD-sfGFP (50 kDa) in the encapsulin and cargo fractions from size-exclusion chromatography. (**C**) Denaturing SDS-PAGE of *in vitro* loaded native cysteine desulfurase (WT CyD) and cysteine desulfurase with the N-terminal domain removed (ΔNTD-CyD) to determine presence of SrpI shell monomer (35 kDa), WT CyD (68 kDa), and ΔNTD-CyD (45 kDa) in the encapsulin and cargo fractions from size-exclusion chromatography. WT CyD is subject to proteolysis, leading to the presence of a ΔNTD-CyD band in the WT CyD cargo fraction.

The online version of this article includes the following figure supplement(s) for figure 4:

**Figure supplement 1.** Analysis of sfGFP and 225NTD-sfGFP loading into SrpI encapsulin.
**Figure supplement 2.** Full Coomassie stained gel for panels A and B of *Figure 4*.
**Figure supplement 3.** Full gel for fluorescent panel of *Figure 4A*.
**Figure supplement 4.** Full gel of the SrpI, ΔNTD-CyD panel from *Figure 4C*.
**Figure supplement 5.** Full gel of the SrpI, WT CyD panel from *Figure 4C*.

## Structural details of the SrpI shell revealed by cryo-EM

Motivated by our negative stain TEM results, we sought to obtain a high-resolution structure of the nanocompartment complex by single-particle cryo-electron microscopy (cryo-EM). All encapsulin structures published thus far have belonged to the Family 1 encapsulins. These structures are all icosahedral, vary in size from 24 to 42 nm in diameter, and have a triangulation number of T = 1, T = 3, or more recently, T = 4 (*Sutter et al., 2008*; *Akita et al., 2007*; *McHugh et al., 2014*; *Giessen et al., 2019*). Cryo-EM analysis was performed on purified holo-SrpI to resolve the shell structure at 2.2 Å resolution. This represents the first Family 2 encapsulin structure and is the highest resolution structure for an encapsulin to date, allowing for accurate atomic model building (*Figure 6A*, *Figure 6—figure supplements 1* and *2*). The SrpI encapsulin is 24.5 nm in diameter and has a T = 1 icosahedral capsid formed by the self-assembly of 60 SrpI monomeric subunits (*Figure 6A*), similar to previously reported encapsulin structures. Given the structural similarity of the entire shell, it is unsurprising that the SrpI monomer also shares the canonical HK97 fold found in Family 1 encapsulins and Caudovirales bacteriophages (*Figure 6B*, *Figure 6—figure supplement 3C*).

Most of the topological elements are shared between the monomeric subunits of encapsulins and Caudovirales shells, including the A-domain (axial domain), E-loop (extended loop), and P-domain (peripheral domain). However, unlike other Family 1 encapsulin shell proteins, the Family 2a SrpI possesses an extended N-terminal arm (N-arm) that is more characteristic of bacteriophage structures (*Figure 6—figure supplement 3C*; *Duda and Teschke, 2019*). Similar to other HK97 bacteriophage capsids, the N-arm of the SrpI shell intertwines with the neighboring subunit to create a chainmail-like topology (*Figure 6—figure supplement 4*). The most striking differences in quaternary structure between the SrpI encapsulin and the previously studied Family 1 encapsulins can be observed at the major vertices that form the 5-fold axis of symmetry. The vertices of the SrpI encapsulin protrude from the capsid yielding a raised 'spike' morphology not found in the Family 1

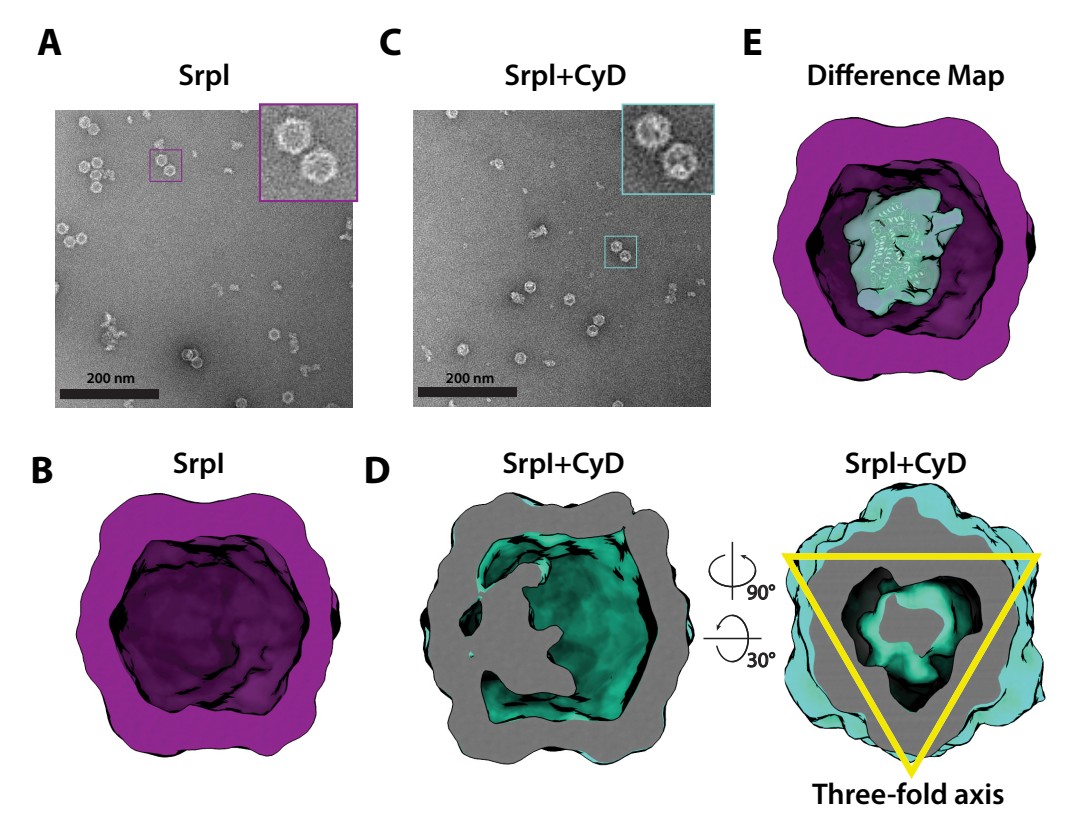

**Figure 5.** Negative stain analysis indicates CyD loading into SrpI encapsulin. (A) Negative stain micrograph of an apo-SrpI shell in contrast with (C) the holo-SrpI shell that includes the CyD cargo. (B and D) 3D reconstruction of apo-SrpI and holo-SrpI, respectively. (E) Difference map showing additional density for the holo-SrpI with a homologous cysteine desulfurase dimer docked in (pdb:6c9e).

encapsulins (*Figure 6—figure supplement 3D*). This difference is due to an extended C-terminus in the SrpI shell protomer that is found near the A-domain, whereas the C-terminus of the Family 1 encapsulins is located farther away from the fivefold axis.

Interestingly, the A-domain of SrpI that forms a pore at the fivefold axis is composed of residues that are positively charged (*Figure 6C*, *Figure 6—figure supplement 5*), in contrast to the negatively charged fivefold axis pore of Family 1 encapsulins (*Giessen et al., 2019*). It is thought that the fivefold pore at the capsid vertices creates a selective barrier to allow encapsulin substrates into the compartment lumen (*Giessen et al., 2019*; *Nichols et al., 2017*; *Sutter et al., 2008*). Cysteine is a likely substrate for the SrpI encapsulin given its cysteine desulfurase cargo enzyme. At physiological pH for *S. elongatus* growing in light, pH 8–8.4 (*Mangan et al., 2016*), roughly 30–54% of free cysteine will have a net charge of −1 and therefore could traverse the positively charged pore exterior (*Cameselle et al., 1986*). While this charge selectivity makes sense in the uniquely high pH environment of *S. elongatus* PCC 7942, this compartment is also found in model bacteria that are known to have a lower cytoplasmic pH such as *Mycobacterium smegmatis* which has been measured to be between pH 6.1 and 7.2 (*Rao et al., 2001*). Because of the wide discrepancy of the cytosolic pH for bacteria that possess the SrpI encapsulin, we sought to understand the sequence conservation of residues at the fivefold pore (*Figure 6—figure supplement 6*). Residues that conferred the positive charge of the pore in *S. elongatus* PCC 7942, Arg 192, and Arg 304, appear not to be well-conserved in other organisms.

In contrast, the residues that appear most conserved are Gly 194 and Pro 196. These residues form the diameter of the inside of the pore. Therefore, it is likely that the size of the pore is also an important constraint for limiting the spectrum of substrates that can enter the compartment (*Williams et al., 2018*). The SrpI encapsulin fivefold pore is 3.7 Å in diameter at its most restrictive

point (*Figure 6C*) as calculated by HOLE (*Smart et al., 1996*). Furthermore, modeling of cysteine in the pore demonstrates it is likely small enough to enter the nanocompartment (*Figure 6D*).

Unfortunately, during processing and classification, the holo-SrpI encapsulin proved nearly indistinguishable from the apo-SrpI control (*Figure 6—figure supplement 1*). The inability to resolve significant portions of the cargo density may be due to low occupancy, or conformational flexibility of the cargo, which disappears at higher resolutions when many particles are averaged together. The inability to fully resolve cargo protein within the nanocompartment has been observed for previously published encapsulin structures (*Sutter et al., 2008*). However, symmetry expansion and focused classification of the holo-SrpI encapsulin, guided by the 225-NTD-sfGFP-SrpI dataset, revealed additional EM density at the interior surface of the shell that was localized to the three-fold symmetry axis (*Figure 6E*). This corroborates our findings from the holo-SrpI structure obtained via negative stain TEM, which also demonstrated cargo interfacing with the shell at the axis of threefold symmetry (*Figure 5D*).

While the cargo EM density was too weak to accurately build an atomic model of the cysteine desulfurase cargo residues, we were able to determine which shell residues likely interact with cargo density based on their proximity to the putative cargo EM density (*Figure 6E*). Of note, shell residues at the threefold axis neighboring the cargo EM density are highly conserved, which suggests that the interaction between the encapsulin cargo and shell may be conserved (*Figure 6—figure supplement 3A,B*). Namely, residues F262, and Y273 are located near the suggested cargo density.

## Encapsulation of CyD modulates enzymatic activity

Finally, we wanted to understand the enzymatic activity of CyD and assess whether encapsulation affects the cargo enzyme. Enzyme activity was monitored via NADH fluorescence using an assay coupling cysteine desulfurase, which produces alanine, to alanine dehydrogenase (*Dos Santos, 2017*). Unencapsulated CyD was active towards a cysteine substrate and exhibited a $k_{cat}$ of $10 \pm 4$ s$^{-1}$ (*Figure 7*). In accordance with our hypothesis that free cysteine could enter the SrpI encapsulin pore, we found that the encapsulated CyD was roughly sevenfold more active than unencapsulated CyD, with a $k_{cat}$ of $67 \pm 5$ s$^{-1}$ (*Figure 7*). Importantly, the N-terminal domain had no effect on catalytic activity, and unloaded SrpI encapsulin did not exhibit any cysteine desulfurase activity. Rate enhancement of cargo enzymes by Family 1 encapsulins has been observed previously and is discussed below (*Rahmanpour and Bugg, 2013*).

## Discussion

Here, we have identified a unique bacterial nanocompartment and established it as a member of a distinct family of encapsulins, which we name Family 2, that have thus far evaded characterization as an organelle-like compartment.

### Structural analysis of SrpI encapsulin reveals a potential role as a prokaryotic nanocompartment

We report here the first high-resolution structure for Family 2 encapsulins and find that it shares the HK97 fold found in the Family 1 encapsulins. While there are many structural similarities between the Family 1 and Family 2 encapsulin shell proteins, there are notable differences in the structural properties of the individual domains within the HK97 fold that likely confer distinct functions. One considerable difference between the two encapsulin families is the nature of the pentameric vertex of the capsid that forms the major pore of the compartment. The fivefold pore is likely crucial to the organellular function of encapsulins, as it provides a selective barrier for the entrance of compartment substrates based on size and charge (*Nichols et al., 2017*; *Williams et al., 2018*; *Giessen et al., 2019*). Structures determined for the Family 1 encapsulins reveal pores that are negatively charged (*Giessen et al., 2019*; *McHugh et al., 2014*; *Sutter et al., 2008*). Many of the Family 1 encapsulins possess cargo that binds iron, such as the IMEF and FLP containing encapsulins that have been demonstrated to be capable of iron storage (*Giessen et al., 2019*; *He et al., 2016*). The negative charge of the Family 1 pores is therefore noteworthy because it may allow passage of the positively charged iron ions to enter the encapsulin lumen.

In the Family 2 encapsulin structure reported here, we find the electrostatic charge of the pore to be positive – opposite to what is observed in Family 1. The positive charge of the Family 2

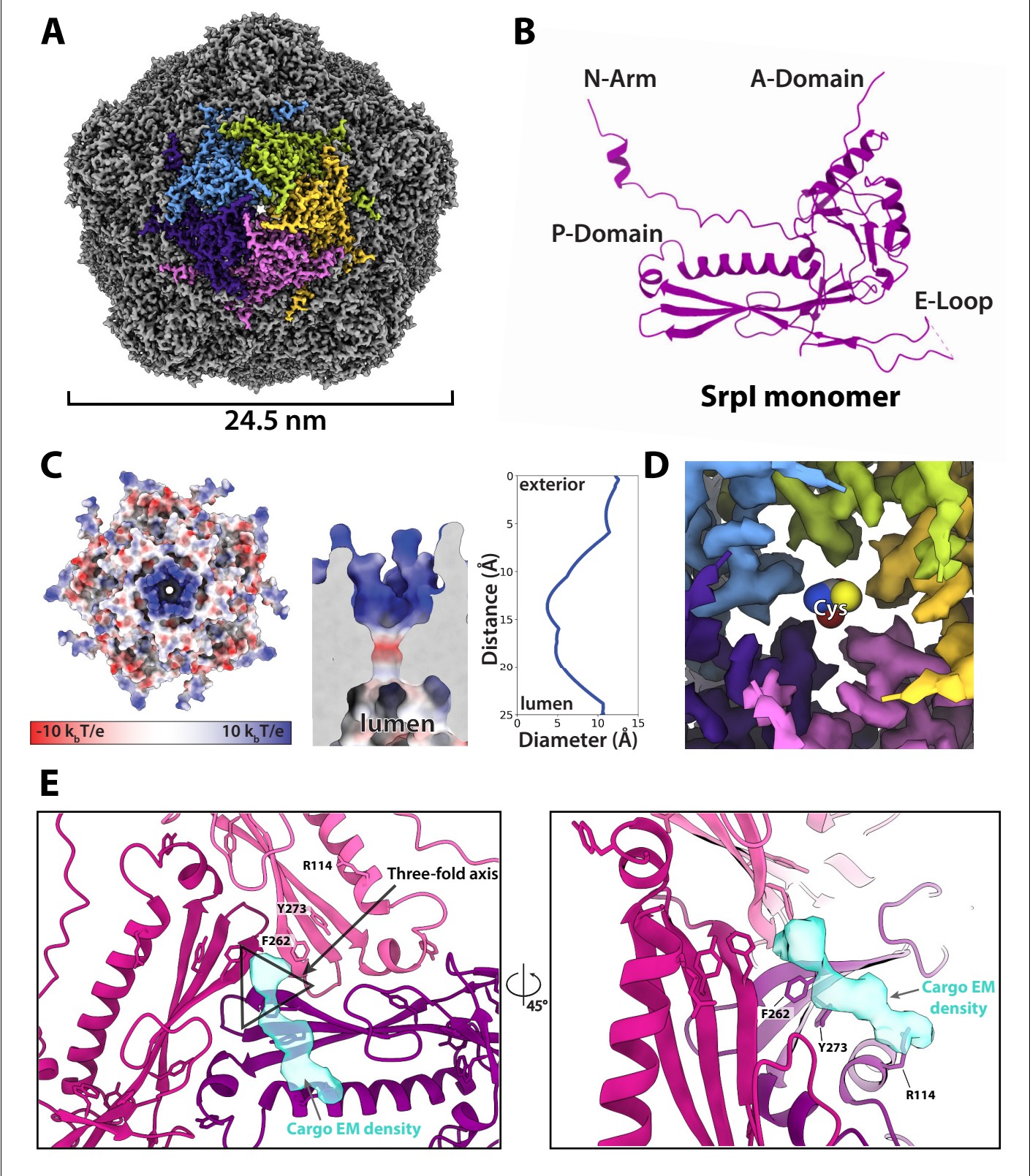

**Figure 6.** CryoEM structure of the SrpI encapsulin reveals a common HK97 fold, a potential mechanism for cysteine selectivity, and a cargo binding site (**A**) The SrpI encapsulin structure at 2.2 Å resolution. This SrpI encapsulin forms a T = 1 icosahedral structure 24.5 nm in diameter. Five subunits around a fivefold axis are shown in distinct colors. (**B**) SrpI monomer subunits have a HK97 fold with the characteristic A-Domain, E-Loop, P-Domain, and

*Figure 6 continued on next page*

*Figure 6 continued*

N-Arm. (**C**) Electrostatic surface potential at the fivefold symmetry axis indicates a relatively neutral pore with an electropositive exterior (left). At its constriction point, the pore is 3.7 Å in diameter. (**D**) Modeling of a cysteine amino acid at the fivefold axis illustrates the possible mechanism of substrate selection (permissivity to cysteines) by the pore. (**E**) Unassigned density (turquoise) near the threefold axis (gray triangle) revealed by symmetry expansion and focused classification of the holo-SrpI cryoEM map shown in two different orientations.

The online version of this article includes the following figure supplement(s) for figure 6:

**Figure supplement 1.** Processing pipeline for the SrpI encapsulin.
**Figure supplement 2.** CryoEM resolution map of SrpI encapsulin.
**Figure supplement 3.** Secondary, tertiary, and quaternary homology between SrpI and other known encapsulins.
**Figure supplement 4.** Chainmail-like topography of SrpI.
**Figure supplement 5.** Electrostatic surface charges at the symmetry axes of the SrpI shell.
**Figure supplement 6.** Conservation of residues at the fivefold pore.

encapsulin is consistent with its likely substrate, cysteine, which will have a net negative charge at physiological pH (*Cameselle et al., 1986*; *Mangan et al., 2016*). This contrast between pore charges suggests an overarching theme that may be shared among the encapsulins: the electrostatic charge of the pore is likely reflective of the charge characteristics of the cargo substrate. Furthermore, the size of the pore also appears to be an important parameter as it selects for the entry of substrate molecules while still maintaining a partitioned barrier from larger molecules in the surrounding environment. Initial work has begun toward dissecting how the size of the pore affects mass transport of substrates varying in size. One such study has engineered the pore of the *T. maritima* encapsulin to allow for the diffusion of metals such as terbium, which is nearly double the atomic radius of the native iron substrate (*Williams et al., 2018*). Further mutational studies of pore residues will be needed to better understand how the properties of encapsulin pores affect permeability and function of nanocompartments both *in vitro* and *in vivo*.

## The effect of encapsulation on cargo protein function

Our enzymatic activity data of the CyD cargo provide evidence that encapsulation of the enzyme is important for its activity. We found that the $k_{cat}$ for the encapsulated CyD was almost sevenfold higher than that of the naked cargo *in vitro*. Others have also reported changes in cargo activity upon encapsulation for the Family 1 encapsulins. One such example is the DyP-type peroxidase-containing encapsulin from *Rhodococcus jostii,* which was shown to exhibit an eightfold higher activity on a lignin substrate compared to the unencapsulated peroxidase (*Rahmanpour and Bugg, 2013*). The biochemical function of the FLP-containing encapsulin is also impeded in the absence of shell protein, as it loses the ability to properly store and mineralize iron (*He et al., 2016*; *McHugh et al., 2014*). The exact mechanism by which encapsulation affects cargo activity remains unknown and may differ for the various cargo types found in encapsulin systems. In the case of the CyD cargo, there are precedents for the enhancement of cysteine desulfurase activity in the presence of accessory proteins. For example, the protein SufE from *E. coli* has been shown to increase the activity of the cysteine desulfurase, SufS, 8- to 50-fold (*Loiseau et al., 2003*; *Outten et al., 2003*). SufE binding stimulates an allosteric change in SufS and enables faster regeneration of the SufS active site by the removal of persulfide from the SufS active site, thereby allowing additional reaction cycles (*Outten et al., 2003*; *Singh et al., 2013*). It is possible that SrpI may be acting as an accessory protein for the activity of its cysteine desulfurase cargo analogous to what has been observed for SufE and SufS. On the other hand, the increase in $k_{cat}$ may be merely incidental and the compartment sequesters CyD for a different reason. For example, if SrpI does act similarly to SufE, then it may accumulate a polysulfide chain (*Ollagnier-de-Choudens et al., 2003*) that may need to be protected from the reducing environment of the cytosol. Another possibility is CyD may be sequestered to prevent it from degrading too much cytosolic cysteine during sulfur starvation.

## SrpI may be linked to the canonical sulfur starvation pathway in cyanobacteria

The physiological response to sulfur starvation in cyanobacteria has been studied for decades, yet much is still unknown about the interplay of the known components of the pathway. Genetic and biochemical approaches to study the sulfur starvation response in *S. elongatus* have shown that

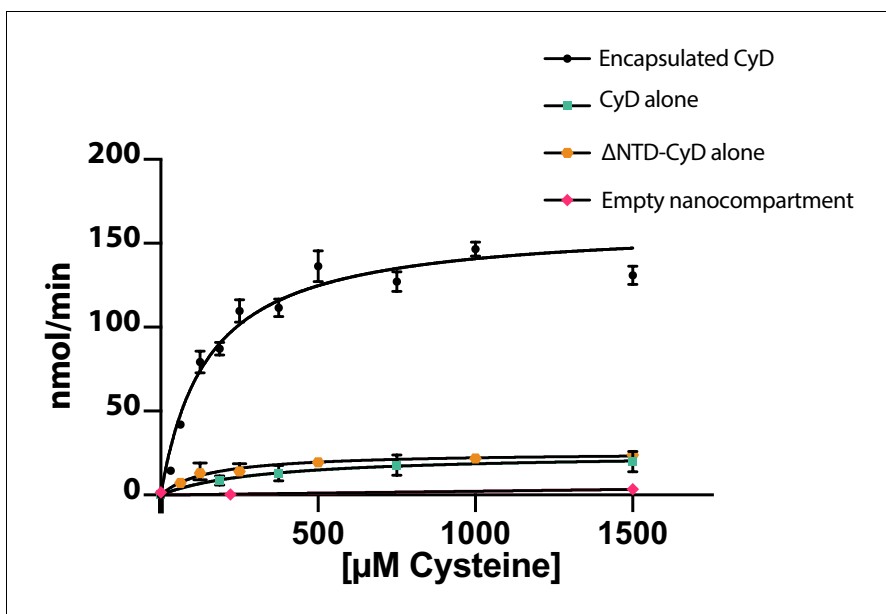

**Figure 7.** Cysteine desulfurase activity is enhanced upon encapsulation. Substrate-dependent activity of encapsulated cysteine desulfurase (encapsulated CyD), unencapsulated cysteine desulfurase (CyD alone), unencapsulated cysteine desulfurase lacking its NTD (ΔNTD-CyD alone), and empty nanocompartment using a coupled-enzyme assay with alanine dehydrogenase and production of NADH as a readout of cysteine desulfurase activity. Error bars are SD of 3–6 replicate experiments. Reactions were carried out at 25°C in 25 mM Tris–HCl pH 8, 150 mM NaCl, 5 mM NAD+, 0.4 Units of alanine dehydrogenase, 200 nM cysteine desulfurase, and varying L-cysteine concentrations.

The online version of this article includes the following figure supplement(s) for figure 7:

**Figure supplement 1.** Ultraviolet-visible absorbance spectra of purified CyD (black) and encapsulated CyD (Red).

---

photosynthesis is halted as phycobiliproteins are disassembled from the thylakoid membrane and proteolyzed by the Clp protease complex to generate free amino acids such as cysteine (*Baier et al., 2014*; *Collier and Grossman, 1992*; *Karradt et al., 2008*). Among the other known responses to sulfur starvation are the upregulation of proteins involved in sulfate transport CysA, CysT, CysW, and SbpA (*Green et al., 1989*; *Laudenbach and Grossman, 1991*). Cyanobacteria are assimilatory sulfate reducers and thus, sulfate is sequentially reduced to sulfite and sulfide by the APS/PAPS pathway, which is then followed by the synthesis of L-cysteine via serine-O-acetyltransferase and cysteine synthase (*Kopriva et al., 2008*; *Schmidt, 1990*; *Schmidt and Christen, 1978*).

Here, we report that *S. elongatus* cells deprived of sulfate dramatically upregulate the SrpI encapsulin and its cysteine desulfurase cargo. Furthermore, it is interesting to note that cyanobacteria that possess SrpI encapsulin genes are found in freshwater or brackish water, but not marine environments (*Supplementary file 4*). Sulfate is often a limiting nutrient in freshwater environments compared to marine habitats; therefore, the presence of SrpI exclusively in freshwater cyanobacteria suggests SrpI could play a role in sulfur starvation response (*Giordano et al., 2005*; *Pilson, 2012*; *Tipping et al., 1998*). Our biochemical characterization of this complex showed it is capable of using free L-cysteine as a substrate. Given that the previously characterized facets of the sulfur starvation pathway have been demonstrated to yield free cysteine, our findings suggest a potential link between the SrpI encapsulin and the rest of the known pathway. The physiological role of the SrpI encapsulin may be elucidated by determining the fate of the sulfide group from cysteine after conversion to alanine by the cysteine desulfurase. For example, the sulfur may be mobilized to a downstream sulfide carrier such as NifU or SufE to be incorporated into iron-sulfur clusters or thiocofactors (*Black and Dos Santos, 2015*). Alternatively, the sulfide from cysteine may remain within the compartment in the form of polysulfide. In this second model, the encapsulin may act as a storage cage for sulfur, similar to how Family 1 encapsulins are thought to act as iron stores (*Giessen et al., 2019*; *He et al., 2016*).

## Homologs of SrpI are found in pathogens

While this report is the first to recognize SrpI as an encapsulin, previous work has observed a role for SrpI homologs in mycobacterial pathogenesis. Previous work on a SrpI homolog from both *M. leprae* and *M. avium* has shown SrpI is the most antigenic protein in human leprosy patients (*Winter et al., 1995*). Because of its ability to elicit a proliferative T-cell response in leprosy patients, SrpI homologs have been proposed as a useful antigen for vaccine development in disease-causing *Mycobacteria* (*Abdellrazeq et al., 2018*; *Abdellrazeq et al., 2020*; *Leite et al., 2015*). In their search for candidate antigens from *M. avium* subsp. paratuberculosis, Leite and colleagues enriched for a complex that was identified as MAP2121c and MAP2120c, the SrpI shell and cysteine desulfurase homologs respectively (*Leite et al., 2015*). Our results here validate the finding that these two proteins share a direct biochemical interaction. Moreover, we have identified the N-terminal domain of the cysteine desulfurase to be essential for its interaction with the shell and have presented structural evidence for the encapsulin shell residues with which the cysteine desulfurase may interact. Hopefully our structural and biochemical characterization of the compartment aids in future studies of the role of SrpI encapsulin in pathogenicity and host immune response.

## The evolutionary origins of encapsulins and the prospect of additional undiscovered families

An evolutionary relationship between the encapsulins and Caudovirales bacteriophages is clear given the shared HK97 fold of the capsid proteins. Exactly how the encapsulins and Caudovirales bacteriophages are related, however, remains an open question (*Koonin and Krupovic, 2018*). It is possible that the HK97 fold derives from a cellular ancestor and was then recruited by a virus-like ancestor of Caudovirales phage (*Krupovic and Koonin, 2017*). Alternatively, the HK97 fold was of viral origin and cellular hosts co-opted the compartments as proteinaceous organelles that enabled some fitness benefit (*Koonin and Krupovic, 2018*; *Krupovic and Koonin, 2017*). These two scenarios may not be mutually exclusive, and it is possible that the ancestry of the HK97 fold is intermingled between the two systems with the interconversion of capsids functioning as Caudovirales phage or prokaryotic encapsulins repeatedly over evolutionary history (*Radford, 2015*). Our identification of SrpI and its homologs as members of an evolutionarily distinct encapsulin family may provide further insights into the divergence and origin of prokaryotic nanocompartments. Already, the breadth and diversity of known encapsulin systems is vast, yet it is likely that more await discovery.

# Materials and methods

## Phylogenetic analysis of encapsulin genes

Homologs of SrpI were compiled using NCBI BLASTp with query sequence WP_011055154.1 (*Synechococcus elongatus*). BLASTp searches were carried out in February 2020 and hits with an E-value <0.01 were collected and used in subsequent phylogenetic analysis. Sequences were aligned using MAFFT v7.453 along with the outgroup sequence WP_004080898.1 (*Thermotoga maritima*). The phylogenetic tree was generated from the MAFFT alignment using IQ-TREE (*Kalyaanamoorthy et al., 2017*) with LG model, four gamma categories and allowing for invariant sites and ultrafast bootstrap with 1000 replicates. Taxonomy metadata for the encapsulin sequences were compiled using the NCBI protein database. Phylogenetic trees were visualized and annotated using 'The Interactive Tree of Life v4' online server (*Letunic and Bork, 2019*).

Genome neighborhood analysis of the Family 2 encapsulin sequences was performed using the Enzyme Function Initiative suite of web tools (*Gerlt, 2017*; *Zallot et al., 2019*). Sequences were compiled with the Enzyme Similarity Tool (ESI) using WP_011055154.1 (*S. elongatus*) as the query sequence. A Uniprot BLAST search was performed using ESI with an E value of 1E-5 and a maximum of 10,000 sequences. The resulting dataset was then submitted to the Genome Neighborhood Tool to identify the 10 genes upstream and downstream of every Family 2 encapsulin hit.

Secondary structure and disorder prediction of the encapsulin-associated cysteine desulfurase, Synpcc7942_B2661, was performed using PsiPred4 and Disopred3 (*Buchan and Jones, 2019*; *Jones and Cozzetto, 2015*). Sequence identity scores for homologs of Synpcc7942_B2661, were determined by aligning sequences using Clustal Omega and analyzing results with Geneius Prime Version 2019.2.1 (*Sievers et al., 2011*). Conserved motifs within the cysteine desulfurase sequence

homologs were determined using the MEME Suite 5.1.1 (*Bailey et al., 2009*). Using Clustal Omega, the compiled cysteine desulfurase sequences were aligned and truncated to only include the N-terminal domain sequence. These truncated sequences were then analyzed using MEME to create sequence logos of the top occurring motifs.

## Molecular cloning, protein expression, and purification

All plasmids were constructed using either Gibson Assembly (NEB) or SLiCE (*Zhang et al., 2012*) homology-based cloning strategies. Each construct was cloned into a pET-14-based destination vector with a T7 promoter. These constructs were transformed into *E. coli* BL21 (DE3) LOBSTR cells for protein expression (*Andersen et al., 2013*). Cells were grown in LB media containing 60 μg/mL kanamycin at 37°C, shaking at 250 rpm to an optical density ($OD_{600}$ = 0.5–0.6) before lowering the temperature to 18°C, inducing with 0.5 mM IPTG, and growing overnight. Liquid cultures were harvested via centrifugation (4000 x *g*, 20 min, 4°C), flash frozen in liquid nitrogen, and stored at −80°C for future use.

All cysteine desulfurase constructs used in enzyme activity and electron microscopy experiments were purified using an N-terminal SUMO tag containing a poly-histidine sequence. Pellets were resuspended in lysis buffer (25 mM Tris-HCl pH 7.5, 150 mM NaCl, 20 mM imidazole) supplemented with 0.1 mg/ml lysozyme and 1 U/mL Benzonase Endonuclease (Millipore Sigma). Sample was lysed with three passages through an Avestin EmulsiFlex-C3 homogenizer and clarified via centrifugation (15,000 x *g*, 30 min, 4°C). The resulting supernatant was then bound to HisPur Ni-NTA resin (Thermo-Fisher Scientific) for 1 hr at 4°C, followed by application of the sample to a gravity column. The resin was then washed with 30 resin volumes of wash buffer (25 mM Tris-HCl pH 7.5, 150 mM NaCl, 40 mM imidazole) prior to eluting with 25 mM Tris-HCl pH 7.5, 150 mM NaCl, 350 mM imidazole. The eluate was then concentrated and desalted into 25 mM Tris-HCl pH 8, 150 mM NaCl using Econo-Pac 10DG desalting columns (Bio-Rad). The SUMO tag was removed by adding SUMO-protease to the purified sample at a 1:200 (protease: purified protein) molar ratio and allowing cleavage overnight at 4°C. The sample was then further purified by size exclusion chromatography using a Superose 6 Increase column (GE Life Sciences) and fractions were analyzed by SDS-PAGE using 4–20% Criterion polyacrylamide gels (Bio-Rad) and visualized with GelCode Blue stain (ThermoFisher).

## *In vitro* loading of cargo into SrpI encapsulin

To obtain sufficient quantities of the SrpI shell for *in vitro* loading experiments, the protein was purified from *E. coli* inclusion bodies. Serendipitously, adding a C-terminal 6X-His tag to SrpI yielded high quantities of insoluble shell protein. Purification, denaturation, and folding of shell protein was performed as previously described (*Palmer and Wingfield, 2012*). Briefly, cell pellets were resuspended in solution buffer (50 mM Tris-HCl pH 8, 1% Triton-X100, 100 mM NaCl, 10 mM DTT, 0.1 mg/ml lysozyme, and 1 U/mL Benzonase Endonuclease) and lysed with an Avestin EmulsiFlex-C3 homogenizer. The lysate was then centrifuged at 11,000 x *g* for 20 min at 4°C, and the resulting pellet was resuspended in washing buffer A (50 mM Tris-HCl pH 8, 0.5% Triton-X100, 100 mM NaCl, 10 mM DTT) followed by sonication and centrifugation at 11,000 x *g* for 20 min, 4°C. The resulting pellet was then resuspended in washing buffer B (50 mM Tris-HCl pH 8, 100 mM NaCl, 10 mM DTT) followed by sonication and centrifugation again at 11,000 x *g* for 20 min, 4°C. The pellet containing SrpI shell protein was then solubilized with extraction buffer (50 mM Tris-HCl pH 7.4, 6M guanidine hydrochloride, 50 mM DTT), flash frozen and stored at −80°C for future use.

Refolding was performed by 100-fold dilution in refolding buffer (50 mM CAPS pH 10, 250 mM arginine, 150 mM NaCl, 20% Glycerol, 10 mM DTT). For *in vitro* cargo loading, refolding was performed by adding cargo protein prior to shell protein in a 10:1, cargo: compartment ratio. Sample was concentrated in an Amicon stirred cell (Millipore Sigma) using a 10 kDa MWCO filter, followed by desalting into 50 mM CAPS pH 10, 250 mM arginine, 150 mM NaCl. Subsequent purification was performed using either a Superose 6 Increase column or a HiPrep 16/60 Sephacryl S-500 HR (GE Life Sciences).

## *S. elongatus* PCC 7942 growth and sulfate deprivation

*S. elongatus* PCC 7942 was grown in BG-11 media (*Allen, 1968*) at 30°C with shaking (185 rpm) under white fluorescent lights at 60–100 μE. After the liquid culture reached log phase

(OD$_{750}$ = 0.4–0.5), sulfate starvation was performed by centrifugation of liquid culture (5000 x *g*, 20 min, 25°C), resuspension of cells in BG-11 media lacking sulfate (*Collier and Grossman, 1992*), and repeated for a total of three washes. Control samples were washed and resuspended using normal BG-11. Samples were then returned to the above growth conditions for continued growth in sulfate dropout BG-11 media. Phycocyanin and chlorophyll levels were quantified by removing 1 mL of culture at predetermined times and measuring 620 nm and 680 nm absorbance levels respectively, normalized to cell density at 750 nm.

## Identification of protein complex upregulated under sulfate starvation

Sulfate-starved and control *S. elongatus* PCC 7942 liquid cultures (50 mL) were harvested via centrifugation (4000 x *g*, 20 min, 4°C), flash-frozen in liquid nitrogen and stored at −80°C for future processing. Pellets were lysed via sonication and clarified by centrifugation (15,000 x g, 20 min, 4°C). Clarified lysates were analyzed using 4–20% Criterion polyacrylamide gels (Bio-Rad) and visualized by silver staining using Pierce Silver Stain Kit (ThermoFisher Scientific). Gel bands were excised and sent to UC Davis Proteomics Core Facility. In-gel proteolytic digestion of the samples was performed followed by LC/MS analysis with a Q Exactive Hybrid Quadrupole-Orbitrap. Spectra were searched against the *S. elongatus* PCC 7942 proteome and analyzed using Scaffold 4.

## Generation of SrpI, SrpI, and CyD knockout strains

DNA constructs were designed by flanking a chloramphenicol resistance gene with 1200 bp homology arms upstream and downstream the locus of interest – SrpI, SrpI and CyD, and neutral site 3 (NS3). *Synechococcus elongatus* PCC 7942 were transformed according to *Golden and Sherman, 1984*. Briefly, cultures were grown to OD750nm = 0.5. Cultures were centrifuged at 18,000 x g for 2 min. Pellets were washed with 100 mM CaCl$_2$ and spun again at 18,000 x g for 2 min. Pellets were resuspended in BG-11 media followed by addition of plasmid and grown for 16 hr in the dark at 30°C. Transformants were then plated onto BG-11 agar plates containing chloramphenicol (10 µg/mL) and placed under 60 µE of light at 30°C. Single colonies were then genotyped by PCR amplification.

## Cysteine desulfurase activity

Cysteine desulfurase activity was performed using a coupled enzyme assay with alanine dehydrogenase purchased from Sigma-Aldrich, product number 73603. Reactions were carried out at 25°C in 25 mM Tris-HCl pH 8, 150 mM NaCl, 5 mM NAD$^+$, 0.4U alanine dehydrogenase, 200 nM cysteine desulfurase, and varying L-cysteine concentrations. Cysteine desulfurase concentration for compartment loaded and non-loaded samples was determined by PLP absorbance at 420 nm (*Figure 7—figure supplement 1*). Activity was monitored by production of NADH using an Infinite M1000 plate reader (Tecan) with excitation at 340 nm and emission at 460 nm. Reactions excluding cysteine desulfurase were used as negative controls for background subtraction. Activity data is reported as the initial rate of product formation over substrate concentration and fitted with the Michaelis–Menten equation using GraphPad Prism 8.

## Negative stain TEM

Holo-SrpI and apo-SrpI samples were diluted to 300 nM in TEM Buffer (50 mM CAPS pH 10, 250 mM arginine, 150 mM NaCl). Four µL of each sample was placed onto a 400-mesh continuous carbon grid that had been glow discharged (Tergeo, Pie Scientific). After adsorption of the sample onto the grid (2 min at room temperature), the sample was stained in five successive rinses with 40 µL droplets Uranyl Formate (UF; 2% w/v in water). To ensure stain was thick, and penetrated the shell interior, the grids sat for 30 s with a droplet of UF. Grids were briefly side-blotted with Whatman filter paper for 1 s, leaving a very thin, but still visible, amount of stain still on the grid, followed by air drying for 10 min. Grids were visualized with an FEI Tecnai F20 electron microscope operating at 120 keV. For each construct,~100 micrographs were collected on a Gatan Ultrascan 4 k CCD camera, at a magnification of 80,000x (1.37 Å/pix) and a defocus range from −0.5 to −2.0 µm defocus.

Each dataset was processed identically using RELION (*Scheres, 2012*). Briefly, CTF estimation was performed using CTFFIND4 (*Rohou and Grigorieff, 2015*), and particles were picked using RELION's built-in LoG autopicker. Roughly 5000 particles were extracted for each dataset, binned fourfold, followed by 2D classification into 20 classes. All classes that resembled particles (~80% of

the initial particles picked), were selected for a final refinement without symmetry imposed using a hollow sphere of 25 nm as a reference. Both apo and holo constructs gave a final resolution estimate of ~18 Å.

## CryoEM sample preparation, data acquisition, and processing

Samples were prepared on UltrAuFoil 1.2/1.3 gold grids (Quantifoil). Grids were initially washed with two drops of chloroform and allowed to air dry. Of note, no glow discharge step was performed. A 2 mg/mL solution of the SrpI encapsulin in TEM Buffer supplemented with 0.05% NP-40 was applied to the grid, and immediately plunge-frozen in liquid ethane using a Vitrobot Mark IV (blot force 5, 3 s blot, 100% humidity, 4°C, 1 s drain time). For microscope and collection parameters see *Supplementary file 6*. Briefly, the holo-SrpI sample was collected on a Titan Krios, and the apo- and sfGFP- samples on Talos Arctica. Data was processed within the RELION pipeline (*Scheres, 2012*), with defocus estimation using CTFFIND4 (*Rohou and Grigorieff, 2015*). Particles were automatically picked with LoG-picker and processed in accordance with *Figure 6—figure supplement 1*.

## Atomic model building and refinement

The final map for holo-SrpI was the highest resolution of all the states, and was therefore used for model building. The final reconstruction was post-processed in RELION following the default protocols with a solvent mask that extended ~5 Å past the reconstruction with a ~ 3 Å gaussian soft-edge (RELION calculated b-factor of −35). An atomic model of one asymmetric unit of the SrpI shell was built into this post-processed map with COOT (*Emsley et al., 2010*), and then refined with nearest neighbors using the real space refinement tool in PHENIX (*Adams et al., 2010*). The MTRIAGE program within PHENIX was used to compute the model vs. map FSC, and HOLE was used to analyze the pores at the fivefold symmetry axis (*Smart et al., 1996*). For the apo-SrpI and 225-NTD-sfGFP-SrpI structures, the shell density was indistinguishable compared to the holo-SrpI, so no model refinement was performed. Instead, these maps were used to guide interpretation of additional density within the shell, and to calculate difference maps between the holo- and apo- states using UCSF Chimera (*Goddard et al., 2018*). All coordinates and maps were visualized with UCSF ChimeraX and Pymol (*Goddard et al., 2018*); The PyMOL Molecular Graphics System, (Version 2.3.2 Shrödinger, LLC).

Additionally, symmetry expansion and focused alignment-free 3D classification were performed with RELION for all states. For symmetry expansion, an asymmetric reconstruction was used as the input. The command relion_particle_symmetry_expand with the flag –sym I1 was used to expand the dataset 60-fold to account for the icosahedral symmetry, and properly reassign the Euler angles in order to place each unique subunit into the same location. Following expansion, a local refinement (C1) was performed to adjust for any slight variations in Euler angles that may exist. A generous mask encompassing a single subunit and extending about 50% of the way through neighboring subunits was used for focused alignment-free classification into eight classes. In order to prevent classification of noisy features, the expectation-step was limited to 20 Å and tau was set to 8. Three classes showed similar additional density in the region depicted in *Figure 6E*, while five classes showed no clear supplemental density beyond the shell itself.

## Acknowledgements

We thank Julia Borden, Cecilia Blikstad, Caleb Cassidy-Amstutz, John Desmarais, Eli Dugan, Avi Flamholz, Evan Groover, Shin Kim, and Thomas Laughlin for their assistance, helpful discussions, and critical reading of the manuscript. We acknowledge the UC Davis Proteomics Core Facility for mass spectrometry data collection and the UC Berkeley Electron Microscope Laboratory for training with TEM. We thank Patricia Grob and Daniel Toso for microscope support and Abhiram Chintangal for computational support. This work was supported by a grant from the US Department of Energy (no. DE-SC00016240) to DFS, BL was supported by an NSF-GRFP grant (no. 1106400), and EN is a Howard Hughes Medical Institute Investigator.

## Additional information

### Funding

| Funder | Grant reference number | Author |
|---|---|---|
| U.S. Department of Energy | Grant DE-SC00016240 | David F Savage |
| National Science Foundation | GRFP-1106400 | Benjamin LaFrance |
| Howard Hughes Medical Institute | | Eva Nogales |

The funders had no role in study design, data collection and interpretation, or the decision to submit the work for publication.

### Author contributions

Robert J Nichols, Conceptualization, Resources, Data curation, Formal analysis, Validation, Investigation, Visualization, Methodology, Writing - original draft, Writing - review and editing; Benjamin LaFrance, Conceptualization, Data curation, Formal analysis, Validation, Methodology, Writing - review and editing; Naiya R Phillips, Amanda J Bischoff, Data curation, Writing - review and editing; Devon R Radford, Investigation; Luke M Oltrogge, Conceptualization, Methodology, Writing - review and editing; Luis E Valentin-Alvarado, Data curation, Formal analysis, Investigation, Methodology, Writing - review and editing; Eva Nogales, Supervision, Funding acquisition, Project administration, Writing - review and editing; David F Savage, Conceptualization, Supervision, Funding acquisition, Project administration, Writing - review and editing

### Author ORCIDs

Robert J Nichols ![ORCID] https://orcid.org/0000-0002-8476-0554
Naiya R Phillips ![ORCID] https://orcid.org/0000-0003-1836-5182
Luke M Oltrogge ![ORCID] http://orcid.org/0000-0001-5716-9980
Eva Nogales ![ORCID] https://orcid.org/0000-0001-9816-3681
David F Savage ![ORCID] https://orcid.org/0000-0003-0042-2257

### Decision letter and Author response

Decision letter https://doi.org/10.7554/eLife.59288.sa1
Author response https://doi.org/10.7554/eLife.59288.sa2

## Additional files

### Supplementary files

• Supplementary file 1. Total counts of Family 1 and Family 2 encapsulins found in prokaryotic genomes. Number of total Family 1 and Family 2 homologs compiled using NCBI BLASTp (E-value <0.01). Accession IDs WP_004080898.1 (*T. maritima* encapsulin shell) and WP_011055154.1 (*S. elongatus* PCC 7942) encapsulin shell genes were used as Family 1 and Family 2 queries, respectively. Results based on NCBI's Genome Information resource (February 2020).

• Supplementary file 2. Genome neighborhood analysis of Family 2a shell genes. Co-occurrence and median gene distance of genes found to neighbor Family 2a shell genes using the EFI-GNT web tool. Open reading frames neighboring Family 2a shell genes in the European Nucleic Acid (ENA) database are grouped by shared pfam values.

• Supplementary file 3. Genome neighborhood analysis of Family 2b shell genes. Co-occurrence and median gene distance of genes found to neighbor Family 2b shell genes using the EFI-GNT web tool. Open reading frames neighboring Family 2b shell genes in the European Nucleic Acid (ENA) database are grouped by shared pfam values.

• Supplementary file 4. Family 2a shell genes are found in freshwater and brackish water cyanobacteria, but not marine cyanobacteria. NCBI BLASTp results of Family 2a shell homologs found in cyanobacteria. Results based on NCBI's Genome Information resource (February 2020). Environment

annotations based on the Joint Genome Institute (JGI) Integrated Microbial Genomes and Micro-biomes (IMG/M) database and (*Shih et al., 2013*).

- Supplementary file 5. Total spectrum count of all identified proteins from liquid chromatography-mass spectrometry of excised high-molecular-weight bands from *Figure 2B*.
- Supplementary file 6. Data collection, reconstruction, and processing statistics.
- Transparent reporting form

## Data availability

Cryo-EM maps of holo and apo-SrpI have been deposited at the EM Data Resource with accession codes EMD-22094 and EMD-22095 respectively. The refined coordinate model has been deposited at the Protein Data Bank (PDB) with accession code 6X8M and 6X8T.

The following datasets were generated:

| Author(s) | Year | Dataset title | Dataset URL | Database and Identifier |
|---|---|---|---|---|
| LaFrance BJ, Nichols RJ, Phillips NR, Oltrogge LM, Valentin-Alvarado LE, Bischoff AJ, Savage DF, Nogales E | 2020 | CryoEM structure of the holo-SrpI encapsulin complex from Synechococcus elongatus PCC 7942 | https://www.rcsb.org/structure/6X8M | RCSB Protein Data Bank, 6X8M |
| LaFrance BJ, Nichols RJ, Phillips NR, Oltrogge LM, Valentin-Alvarado LE, Bischoff AJ, Savage DF, Nogales E | 2020 | CryoEM structure of the apo-SrpI encapasulin complex from Synechococcus elongatus PCC 7942 | https://www.rcsb.org/structure/6X8T | RCSB Protein Data Bank, 6X8T |
| LaFrance BJ, Nichols RJ, Phillips NR, Oltrogge LM, Valentin-Alvarado LE, Bischoff AJ, Savage DF, Nogales E | 2020 | CryoEM structure of the holo-SrpI encapsulin complex from Synechococcus elongatus PCC 7942 | https://www.ebi.ac.uk/pdbe/entry/emdb/EMD-22094 | Electron Microscopy Data Bank, 22094 |
| LaFrance BJ, Nichols RJ, Phillips NR, Oltrogge LM, Valentin-Alvarado LE, Bischoff AJ, Savage DF, Nogales E | 2020 | CryoEM structure of the apo-SrpI encapasulin complex from Synechococcus elongatus PCC 7942 | https://www.ebi.ac.uk/pdbe/entry/emdb/EMD-22095 | Electron Microscopy Data Bank, 22095 |
| Nichols RJ, LaFrance BJ, Phillips NR, Oltrogge LM, Valentin-Alvarado LE, Bischoff AJ, Nogales E, Savage DF | 2020 | CryoEM SPA of Holo-SrpI Encapsulin Complex (Raw Frames) | https://www.ebi.ac.uk/pdbe/emdb/empiar/entry/10506/ | Electron Microscopy Public Image Archive, 10506 |
| Nichols RJ, LaFrance BJ, Phillips NR, Oltrogge LM, Valentin-Alvarado LE, Bischoff AJ, Nogales E, Savage DF | 2020 | CryoEM SPA of Apo-SrpI Encapsulin Complex (Raw Frames) | https://www.ebi.ac.uk/pdbe/emdb/empiar/entry/10510/ | Electron Microscopy Public Image Archive, 10510 |

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
