## [Decision Letter]

**Acceptance summary:**

In this study, Nichols et al. use a combination of biochemical and electron microscopic experiments to characterize a unique family of encapsulin nanocompartments in a prokaryotic cell. The study reveals structural determinants that likely confer family-specific substrate selectivity and suggests an evolutionary link between encapsulins and Caudovirales bacteriophages. The work represents a significant contribution to the body of literature on encapsulin systems and is of wider interest to those interested in the physiology and ecology of sulfate metabolism in bacteria.

**Decision letter after peer review:**

Thank you for submitting your article "Discovery and characterization of a novel family of prokaryotic nanocompartments involved in sulfur metabolism" for consideration by *eLife*. Your article has been reviewed by 3 peer reviewers, and the evaluation has been overseen by Sriram Subramaniam as Reviewing Editor and Gisela Storz as the Senior Editor. The following individual involved in review of your submission has agreed to reveal their identity: Jon Marles-Wright (Reviewer #3).

The reviewers have discussed the reviews with one another and the Reviewing Editor has drafted this decision to help you prepare a revised submission.

Summary:

In this study Nichols et al., describe a detailed characterization of a unique family of encapsulin nanocompartments formed by SrpI shell protein in Synechococcus elongatus. The nanocompartment is conserved across bacterial phyla, is upregulated upon sulfur starvation, and encapsulates cysteine desulfurase (CyD) as a cargo enzyme. The molecular determinants of CyD encapsulation and its interactions with SrpI shell protein were further investigated by in-vitro experiments, negative stain TEM, and high-resolution structure determination of the noncompartment using cryo-EM. The study reveals structural determinants that likely confer family-specific substrate selectivity and suggests an evolutionary link between encapsulins and Caudovirales bacteriophages. The work represents a significant contribution to the body of literature on encapsulin systems and is of wider interest to those interested in the physiology and ecology of sulfate metabolism in bacteria.

Essential revisions:

The authors have used "organelle" to describe SrpI encapsulins in some instances in the manuscript. It may be more appropriate replace it with "organelle-like" compartment/structure.

There is a lack of in vivo experimentation in S. elongatus (the host) that should be done with such an important/novel finding; especially given the expertise the Savage lab has with S. elongatus. Most importantly, what happens in a SrpI deletion mutant? What happens when you delete or overexpress CyD in S. elongatus with or without SrpI, with or without sulfur depletion. Are nanocompartments observed in TEM of wt S. elongatus that disappear in an SprI mutant? What is the compartment copy number in wt S. elongatus per cell? The findings suggest it should be roughly equivalent to enzyme copy number?

One thing of particular note is the extensive NTD encapsulation domain that the cysteine desulfurase possesses. It would be interesting to see more made of this in the results, especially in the context of the much shorter encapsulation sequences seen in the type 1 encapsulins. Given the unusual amino acid composition of this domain – to me it looks like a model rigid linker with the high proportion of proline residues interspersed with T/S – it may be more structured than predicted. Some bacteriophage proteins with HK97-like folds have an accessory protein that acts as a molecular staple. Could you comment on whether this domain may act in this way? Did you assay the Δ-NTD version of the cysteine desulfurase to see if this domain influences the enzymatic activity?

---

## [Author Response]

Essential revisions:The authors have used "organelle" to describe SrpI encapsulins in some instances in the manuscript. It may be more appropriate replace it with "organelle-like" compartment/structure.

We have changed all mentions of organelle to explicitly refer to the compartment as organelle-like to avoid confusion.

There is a lack of in vivo experimentation in S. elongatus (the host) that should be done with such an important/novel finding; especially given the expertise the Savage lab has with S. elongatus. Most importantly, what happens in a SrpI deletion mutant? What happens when you delete or overexpress CyD in S. elongatus with or without SrpI, with or without sulfur depletion. Are nanocompartments observed in TEM of wt S. elongatus that disappear in an SprI mutant? What is the compartment copy number in wt S. elongatus per cell? The findings suggest it should be roughly equivalent to enzyme copy number?

We agree that in vivo data is very important for the presented finding. We have, in fact, attempted several in vivo experiments in *S. elongatus* PCC 7942. We have generated mutant strains of *S. elongatus* for which we have made deletions of the SrpI encapsulin and cargo protein including: SrpI shell knockout and SrpI shell / CyD knockout. Although there is not a cohesive conclusion from these particular experiments, we hope that the strains and results may enable future studies seeking to address the physiological function of the SrpI encapsulins.

Additional figures include Supplementary Figure 2-2 A and B in which we sought to obtain a growth phenotype for the SrpI and CyD knockout. In Supplementary Figure 2-2 A we did not find a growth defect for any of the mutants using nutrient replete BG-11 growth medium. In Supplementary Figure 2-2B, we show that there is no growth defect in *S. elongatus* mutants lacking the SrpI encapsulin genes upon sulfate starvation.

We have not observed nanocompartments in TEM experiments examining our strain under nutrient replete conditions. We expect due to their small size and likely low contrast, the nanocompartments would be difficult to detect within the context of the cytoplasmic environment using negative stain TEM.

In text changes:

“After 48 hours of sulfur starvation, the top hits, as determined by total spectral counts, were the putative encapsulin shell protein, SrpI, and the product of the neighboring gene (Synpcc7942_B2661), a cysteine desulfurase (Figure 2C). While we were able to detect upregulation of SrpI and cysteine desulfurase during sulfur starvation, knockout mutants for these genes in *S. elongatus* PCC 7942 did not yield a growth defect phenotype under nutrient replete nor sulfur starvation conditions (Supp. Figure 2-2)”.

(Materials and methods)

**“**Generation of SrpI, SrpI and CyD knockout strains

DNA constructs were designed by flanking a chloramphenicol resistance gene with 1200bp homology arms upstream and downstream the locus of interest – SrpI, SrpI and CyD, and neutral site 3 (NS3). […] Transformants were then plated onto BG-11 agar plates containing chloramphenicol (10ug/mL) and placed under 60uE of light at 30°C. Single colonies were then genotyped by PCR amplification.”

One thing of particular note is the extensive NTD encapsulation domain that the cysteine desulfurase possesses. It would be interesting to see more made of this in the results, especially in the context of the much shorter encapsulation sequences seen in the type 1 encapsulins. Given the unusual amino acid composition of this domain – to me it looks like a model rigid linker with the high proportion of proline residues interspersed with T/S – it may be more structured than predicted. Some bacteriophage proteins with HK97-like folds have an accessory protein that acts as a molecular staple. Could you comment on whether this domain may act in this way? Did you assay the Δ-NTD version of the cysteine desulfurase to see if this domain influences the enzymatic activity?

This is an interesting idea, however, we do not have any evidence that the NTD of CyD acts in this way. A new study has recently dissected the interaction between the encapsulation sequence of the cargo and the shell interior for the Family 1 encapsulins (Altenburg et al. 2021). We hope that similar exhaustive structural and biochemical studies will be performed on the Family 2 encapsulins that were identified in this paper.

We did test the Δ-NTD CyD activity alongside the native CyD and encapsulated CyD, and found that the NTD did not significantly change the enzymatic activity. Figure 7 has been modified to include the enzymatic activity of Δ-NTD CyD.

In-text changes:

“Importantly, the N-terminal domain had no effect on catalytic activity, and unloaded SrpI encapsulin did not exhibit any cysteine desulfurase activity.”

Figure 7 legend: “unencapsulated cysteine desulfurase (CyD alone), unencapsulated cysteine desulfurase lacking its NTD (ΔNTD-CyD alone), and empty nanocompartment”.